# High Dose Intravenous Fish Oil Reduces Inflammation—A Retrospective Tale from Two Centers

**DOI:** 10.3390/nu12092865

**Published:** 2020-09-19

**Authors:** Stanislaw Klek, Dorota Mankowska-Wierzbicka, Lucyna Scislo, Elzbieta Walewska, Magdalena Pietka, Kinga Szczepanek

**Affiliations:** 1General Surgery Unit, Stanley Dudrick’s Memorial Hospital, 15 Tyniecka Street, 32-050 Skawina, Poland; klek@poczta.onet.pl (M.P.); kingaszm@interia.pl (K.S.); 2Department of Gastroenterology, Metabolic Diseases, Internal Medicine and Dietetics, Poznan University of Medical Sciences, 60-001 Poznan, Poland; dmankowska.wierzbicka@gmail.com; 3Department of Clinical Nursing, Institute of Nursing and Midwifery, Faculty of Health Care, Jagiellonian University, 30-130 Krakow, Poland; lscislo@poczta.onet.pl (L.S.); elwalewska@interia.pl (E.W.)

**Keywords:** lipid emulsions, fish oil, omega-3 PUFA, omega-3 fatty acids, parenteral nutrition

## Abstract

Aim: Patients on parenteral nutrition (PN) are prone to inflammation. This may aggravate an existing proinflammatory state and become a critical factor in the development of liver dysfunction (LD). Intravenous fish oil may attenuate this inflammatory state, but data on its use in adults are scarce. The aim of this study was to investigate the effects of adding a pure fish oil intravenous lipid emulsion (ILE) into short- and long-term PN in patients either at risk of, or with existing, inflammation. Methods: A retrospective analysis of 61 patients (32 female, 29 male, mean age 51.5 ± 12.6 years) who received all-in-one PN, including amino acids, glucose, and lipids supplemented with pure fish oil ILE, was performed. Pure fish oil ILE (Omegaven®, Fresenius Kabi, Bad Homburg, Germany) was used along with the standard ILE to reach a fish oil dose of 0.4–0.5 g fish oil/kg/d. Diagnoses were chronic intestinal failure (CIF, *n* = 20), Crohn’s disease (CD, *n* = 22), and ulcerative colitis (UC, *n* = 19). The observation period was 12 months for CIF and 21 days for UC and CD. Results: A reduction in inflammation was noticeable in all patients and became statistically significant in CD (hsCRP *p* < 0.0001, ESR *p* = 0.0034, procalcitonin *p* = 0.0014, Il-6 *p* = 0.001) and UC groups (hsCRP and ESR *p* < 0.0001, Il-6 *p* = 0.0001, TNF-α *p* = 0.0113). In the CIF group, the total bilirubin concentration (*p* = 0.2157) and aspartate transaminase SGOT (*p* = 0.1785) did not vary over time. Conclusions: PN with pure fish oil ILE reduces some inflammatory parameters in IBD and maintains liver function parameters in CIF patients. Fish oil might become a valuable ingredient in both short- and long-term PN in patients at risk of liver dysfunction.

## 1. Introduction

Intravenous lipid emulsions (ILEs) are an integral component of parenteral nutrition (PN) because they are a good source of non-protein energy, as well as essential fatty acids (EFAs) [1,2,3,4]. The first generation of ILEs was derived from soybean oil with a predominance of omega-6 long-chain triglycerides (LCTs). LCTs are effective in terms of supplying energy and provision of EFAs, but due to concerns that an excessive supply of omega-6polyunsaturated (PUFA) might be pro-inflammatory and immunosuppressive, more complex blends of lipid emulsions were developed using mixtures of lipid sources. Therefore, first generation ILEs evolved over the years to second and third generation products [3,4,5,6,7,8,9,10]. The latter include mixtures of medium and long-chain triglycerides (MCT/LCT), olive oil/soybean oil (OO/SO; ClinOleic^®^, Baxter Healthcare, Chicago, IL, USA), SO, MCT, OO and fish oil (FO) (SMOFlipid^®^, Fresenius Kabi, Bad Homburg, Germany), MCT/LCT (SO)/omega-3-acid triglycerides (Lipoplus^®^, B Braun, Melsungen, Germany) and a pure fish oil emulsion (FO, Omegaven^®^, Fresenius Kabi) [6,7,8,9,10,11].

Over the last two decades, a growing body of evidence has emerged that supports the immunomodulatory, anti-inflammatory, and inflammation resolution properties of FO [10,11,12,13,14,15]. A wide range of patient groups can benefit from ILEs containing omega-3 fatty acids, including surgical, cancer, and critically ill patients [12]. In addition, studies in the pediatric population demonstrated liver-protective attributes of FO. In infants and children with intestinal failure-associated liver disease (IFALD), significant improvements in liver function, including a decrease in bilirubin levels and disappearance of jaundice, were seen [16,17,18,19,20,21]. Clinical effects in adult patients were observed within a dose range of 0.1–0.2 g fish oil/kg BW/d, corresponding to lipid doses ranging from 0.6 g/kg BW/d to 1.5 g/kg BW/d. There is still uncertainty about the optimal fish oil dose, but the range of doses that have been observed to be effective can all be achieved with commercially available fish oil-containing products.

Only a few studies have been published in recent years in the present patient populations, but the authors were able to demonstrate advantages of using mixed ILEs with FO [21,22,23,24,25].

The aim of the present study was to analyze the effects of PN supplemented with pure FO-ILE added to standard ILE in two patient groups at particularly high risk of liver complications due to either persistent inflammation in patients with inflammatory bowel disease (IBD) in need of short-term PN or patients with chronic intestinal failure (CIF) on long-term PN. The primary hypothesis was that FO supplementation would help resolve inflammation in IBD and CIF patients. The secondary hypothesis was that the addition of high dose FO, up to 0.5 g fish oil/kg/d, to the PN admixture improves liver function and decreases total bilirubin, in both short-term and long-term PN.

## 2. Methods

A retrospective analysis of patients treated at two centers: Skawina (CIF and HPN center) and Poznan (Gastroenterology Department) between January 2018 and April 2020 was performed. Each patient received PN-containing amino acids, glucose, and standard lipid emulsion without FO with the addition of pure FO-ILE (Omegaven, Fresenius Kabi, Bad Homburg, Germany). Inclusion criteria for both CIF and IBD group were: ≥18 years of age, CIF on PN including lipids (for the CIF group) for at least 6 months, active inflammatory bowel disease (for IBD group), and metabolic stability (the absence of pathological laboratory tests resulting in a change of the PN regime for at least three days (IBD group) or one month (for CIF)).

Exclusion criteria for both groups were: patients with a history of cancer and anti-cancer treatment within the last 5 years, severe hyperlipidemia, severe coagulopathy, severe renal insufficiency, acute thromboembolic events, positive test for HIV, Hepatitis B or C (from medical history), known or suspected drug or alcohol abuse, participation in another interventional clinical trial in parallel or within three months prior to the start of this clinical trial, women of childbearing potential (i.e., females who are not chemically or surgically sterile or females who are not postmenopausal) that have tested positive on a standard pregnancy test (urine dipstick), or/and lactation, allergy to fish oil, or previous treatment with fish oil.

Additional inclusion criteria for CIF patients were: preexisting liver dysfunction defined as an elevation of aspartate transaminase (SGOT), alanine transaminase (SGTP), total bilirubin, gamma-glutamyl transferase (GGTP), and/or alkaline phosphatase (AP) of more than four times the upper level of normal (ULN). Additional inclusion criteria for IBD patients were unchanged status of the severity of the disease for more than 72 h prior to treatment with FO.

ILEs administered during the study period as basic sources of lipids included:-Soybean oil emulsion (SO) (Intralipid, Fresenius Kabi, Germany)-MCT/LCT 50:50 (Lipofundin MCT/LCT, B Braun, Germany)-OO/SO 80:20 (ClinOleic, Baxter Healthcare, Chicago, IL, USA)

The lipid dose in the standard emulsion (without Omega-3-IVLE) did not exceed 0.9 g/kg/day. The long-chain triglyceride dose varied between 0.2 and 0.4 g/kg/day.

Twenty patients were diagnosed with CIF and 41 with IBD; thereof, 22 suffered from Crohn’s disease (CD) and 19 from ulcerative colitis (UC).

The treatment period was 12 months for CIF and 21 days for UC and CD. The treatment period was defined as the time from the first administration of study drug until the final infusion, while the follow up period included four weeks after the last administration of FO-ILE. PN admixtures were prepared in hospital pharmacies.

The study was terminated in the case of any of the following:Serum triglyceride concentration >3 mmol/L (>262.5 mg/dL);intolerable adverse event or serious adverse event (AE);failure of therapeutic safety or tolerability causing an unacceptable risk/benefit ratio.

## 3. Clinical Management

ILEs were infused via a central venous catheter (CVC) as part of all-in-one solutions along with amino acids, glucose, electrolytes, vitamins, and trace elements. For each patient, the required nutritional supply (25–30 kcal/kg ideal body weight/day, 0.9–1.25 g protein/kg body weight/day) was calculated per day. Intravenous fish oil (FO)-ILE was added to the admixture to achieve the provision of 0.4–0.5 g FO/kg/d. CIF patients were allowed to consume food orally or enterally, however the oral provision of nutrients did not exceed 30% of the whole energy and protein intake, which was assessed by a dietitian. IBD patients were allowed to consume fluids and oral nutritional supplements, but the oral provision of nutrients did not exceed 20% of the whole energy and protein intake.

### 3.1. Methodology

Biochemistry, hematology, coagulation, inflammatory parameters, and vital signs were determined. All adverse events were documented and evaluated. Laboratory variables included:-biochemistry: triglycerides, total cholesterol, alkaline phosphatase (AP), aspartate transaminase (SGOT), gamma-glutamyltransferase (GGTP), alanine transaminase (SGTP), sodium, potassium, chloride, magnesium, calcium, phosphate, total bilirubin, creatinine, urea, glucose, albumin, total protein, C-reactive protein (CRP, in CIF), (high-sensitivity CRP, (hsCRP, in IBD), erythrocyte sedimentation rate (ESR), procalcitonin, TNF-α, Interleukine ( Il) -6, Il-10-hematology: leucocytes, platelets, erythrocytes, hemoglobin, hematocrit-coagulation: international normalized ratio (INR)

### 3.2. Clinical Variables

-incidence of cholestasis, defined as either total bilirubin above three-fold the upper limit of normal (ULN) and either gamma-GT or AP above four-fold the ULN, or conjugated bilirubin alone was >2 mg/dL without an explanation for another etiology, e.g., viral hepatitis-adverse events (AEs)-vital signs (blood pressure (mmHg), heart rate (beats/min), body temperature (°C))

Laboratory safety variables and vital signs were recorded:-every three months in CIF patients-every three days in UC and CD patients

AEs were registered during the whole study period and during four weeks of follow-up.

An AE was defined as any untoward medical occurrence in a patient that did not necessarily have a causal relationship with the treatment. A serious AE was any untoward medical occurrence that at any dose, resulted in death, was life-threatening, required inpatient hospitalization or prolongation of existing hospitalization, resulted in persistent or significant disability/incapacity, or was a congenital anomaly/birth defect.

The intensity of all AEs was rated according to the CTCAE v3.0 criteria. Each AE was classified in one of the five following categories, which represented the maximum intensity reported during the evaluation period in question.

-Grade 1 mild-Grade 2 moderate-Grade 3 severe-Grade 4 life-threatening or disabling-Grade 5 death related to AE

### 3.3. Ethics and Consent

The Ethics Committee of the Skawina Hospital approved the study. Patients were approached and enrolled by one of two investigators (SK, DMW). Informed written consent was obtained from each participant before enrollment. The study was carried out following the international ethical recommendations stated in the Helsinki Declaration and was registered at ClinicalTrials.gov under the number: NCT04467138.

### 3.4. Statistical Analysis and Sample Size Calculation

CIF group: all data were analyzed using STATISTICA v.12 (StatSoft Inc., Tulsa, OK, USA). The results are presented as either mean ± standard deviation (SD) or median and interquartile range (IQR) for continuous variables and the number and proportion of patients for categorical variables. Categorical variables were analyzed using the chi-square test of independence. The Shapiro–Wilk test was used to check for normal distribution of data. Non-normally distributed quantitative variables were analyzed using the Kruskal–Wallis test; normally distributed variables were evaluated using analysis of variance (ANOVA). Post hoc analyses were performed using the Tukey test and the Dunn test. Results were considered statistically significant if the *p*-value was less than 0.05.

IBD group: the data are presented as mean ± SD. The normality of the distribution was checked by the Shapiro–Wilk test. Comparisons between the groups were assessed using the Wilcoxon rank-sum test and the paired t-test (for data with normal distribution) to analyze the statistical differences between the variables before and after the intervention. Taking into account normality of the data for testing intergroup significance, either the Kruskal–Wallis test or the one-way ANOVA with post-hoc test was used. The statistical significance level was 0.05. All calculations were performed using STATISTICA v.12 (StatSoft Inc., Tulsa, OK, USA).

## 4. Results

Sixty-one patients (32 female, 29 male, mean age 51.5 ± 12.6 years) were analyzed. All CD (*n* = 22) and UC patients (*n* = 19) completed the 21 day treatment period, and all CIF patients (*n* = 20) completed 12 months of PN + FO treatment.

Table 1 and Table 2 present the changes in liver function tests in CIF patients (Table 1) and the inflammatory status parameters in IBD patients (Table 2). The inflammation parameters improved in CIF patients during the study period. Changes were statistically insignificant.

No adverse events, including sepsis or hypo-/hyperglycemia, were observed during the whole study period.

Patients with IBD were treated with anti-inflammatory medications and corticosteroids, all of them were hospitalized during the study period.

In patients with CD, hsCRP, ESR, procalcitonin, WBC, and IL-6 showed significant improvements, while in patients with UC, findings were significant for hsCRP, ESR, TNF-α, and IL-6.

In CIF patients, SGPT, CRP, and total bilirubin improved over the study period. Yet, differences between the start of the study and after 12 months were not statistically significant (*p* > 0.05 for all liver tests and for CRP).

Authors did not observe any significant differences in the following parameters: triglycerides, total cholesterol, alkaline phosphatase (AP), sodium, potassium, chloride, magnesium, calcium, phosphate, total bilirubin, creatinine, urea, glucose, albumin, and total protein.

## 5. Discussion

Since IFALD can become a life-threatening complication, there is ongoing discussion about the type of ILE that is most beneficial in patients dependent on PN [26,27,28,29,30].

Patients with persistent inflammation or those in need of long-term PN are at particularly high risk of developing liver dysfunction. The most common risk factors for IFALD include sepsis, inflammation, excess energy provision, and some ILE [13]. It is already commonly accepted that the long-term administration of omega-6 PUFAs derived from soybean oil as a sole source of lipids at a dose of >1 g fat/kg body weight/d is a contributing factor in the etiology of IFALD [28]. When reducing energy intake and/or lipid dose fail to be effective in the prevention and/or treatment of IFALD, a change to an ILE with a less pro-inflammatory FA profile, such as FO-containing ILE, may be a valuable option to improve liver function [2,28].

In adults, evidence on the hepatoprotective effects of FO-containing ILEs during long-term PN is scarce. In 2013, a multicenter, randomized controlled clinical trial of the use of PN with FO in 73 adults with stable intestinal failure was published [10]. The observation period was four weeks. The study group receiving a mixed ILE with FO was compared to controls receiving LCT ILE. Mean concentrations of SGTP, SGOT, and total bilirubin, whilst remaining within the reference range were significantly lower with the mixed ILE including FO after the treatment period compared to controls. The mixed ILE with FO also helped to reduce serious adverse events [10]. Another study by Klek et al., published in 2018, was the first to directly compare four ILEs for a 12-month observation period [22]. It showed that liver tests normalize with time regardless of the type of ILE, with soybean oil ILE <1 g/kg/d not proving worse than mixed ILE with FO or other types of ILE (MCT/LCT, OO/SO). The commercially available pure FO ILE (Omegaven^®^, Fresenius Kabi, Germany) was not included in the study.

In some studies on adult hospitalized patients, it has been shown that FO-containing PN had the most favorable effects on clinical outcomes when administered at doses higher than 0.1–0.2 g fish oil/kg b.w./day [25]. This effect has never been confirmed in patient populations other than critically ill or surgical patients. Studies on CIF did not show any positive outcome at that dose. Therefore, the goal of our present study was to assess the impact of PN supplemented with a dose of 0.4–0.5 g/kg/d of pure FO-ILE in addition to a standard lipid emulsion for 12 months in adults with CIF and for 21 days in adult patients with IBD. SGPT and total bilirubin improved in adults with CIF over the study period, but the decrease was not significant. There are two possible explanations to this observation. Firstly, the study group did not suffer from IFALD and most studies have shown good potential of FO in patients with IFALD. Secondly, we exposed patients to a relatively short observation period (12 months). In patients with IBD, we demonstrated that the addition of pure FO ILE to PN significantly improved a number of inflammatory markers, as well as some liver parameters. The beneficial effects on inflammatory status were most pronounced for hsCRP and for IL-6, which were significantly lower at 3 weeks versus baseline values in the CD and in the UC group, respectively.

A few reports of the anti-inflammatory and hepatoprotective effects of pure FO ILE in PN have previously been published. In 2013, Burns et al. published the first reported case of an adult on HPN with IFALD at risk of hepatic failure in the U.S. whose condition improved after the switch to a dose of 1 g/kg/d pure FO ILE as a lipid source in PN [31]. Another case report by Pastor-Clerigues et al. from Spain reported findings of two patients with severe IFALD on pure FO ILE, infused as 1 g FO/kg (5 PN/week) as lipid monotherapy in PN [32]. After the first month, inflammatory, profibrotic, and clinical signs of IFLAD reversed. The effect was maintained over the entire four-month treatment course but vanished with the reintroduction of conventional ILEs. A recent report from Korea showed the reversal of cholestasis after 44 and 59 days, respectively, by increasing the FO content of PN in two adults with short bowel syndrome (SBS) [33]. In these patients, the average daily amount of pure FO ILE ranged from 0.14–0.15 g FO/kg b.w./day, administered either as a sole lipid source or in addition to a mixed ILE with FO (total FO dose of approx. 0.25–0.3 g/kg/d). In a retrospective study from Spain, increasing the amount of FO in PN by means of a combination of a mixed ILE with FO and a pure FO ILE resulted in a higher and significant decrease (*p* < 0.05) in GGT, AP, and triglycerides after an average PN period of 16–17 days compared to patients receiving standard ILE [34].

According to the experts’ consensus from the International Lipid Summit, based on the reduced risk of hepatic injury, FO-containing ILEs are to be preferred over ILEs derived exclusively from soybean oil for adult home PN patients at risk of liver complications [30].

We are aware of some limitations of our study, such as the retrospective approach, the small number of patients included, and the lack of the control group. Nonetheless, in our patient groups, we were able to demonstrate beneficial effects of PN supplemented with a pure FO ILE on inflammatory status in patients with IBD. We may, thus, hypothesize that FO-containing PN might be advantageous in terms of reducing inflammation in patients with IBD at risk of liver dysfunction. Further randomized, controlled trials are necessary to prove this hypothesis.

## Figures and Tables

**Table 1 nutrients-12-02865-t001:** Chronic intestinal failure (CIF) patients—liver tests.

Median (IQR) *	Start of FO-ILE (*n* = 20)	After 12 Months (*n* = 20)	*p*-Value
SGPT (U/L)	28.9 (17–41)	22.8 (16–42)	0.4674
SGOT (U/ll)	19.9 (15–23)	20.4 (17.1–26)	0.1785
GGTP (U/L)	49.5 (21–79)	48.7 (23–155	0.5516
AP (U/L)	147 (108–219)	196 (121–318)	0.5661
Total bilirubin (µmol/L)	12.7 (4.9–17.4)	8.9 (5.4–13.4)	0.2157
C-reactive protein (mg/L)	8.3 (1.7–21.5)	6.8 (0.8–19.5)	0.1778

* IQR = interquadrile range.

**Table 2 nutrients-12-02865-t002:** Inflammatory bowel disease (IBD) patients—changes in inflammatory status during treatment.

	Crohn’s Disease			Ulcerative Colitis		
Parameter	Baseline (*n* = 19)	After Treatment (*n* = 19)	*p*-Value	Baseline (*n* = 22)	After Treatment (*n* = 22)	*p*-Value
hsCRP (mg/L)	80.23	10.84	<0.0001	67.94	9.78	<0.0001
ESR (mm/h)	49.82	22.00	0.0034	66.89	16.42	<0.0001
Procalcitonin (ng/mL)	0.25	0.052	0.0014	0.12	0.05	0.2852
WBC	11.45	7.68	0.0068	11.05	8.58	0.1138
TNF-α (pg/mL)	0.13	0.085	0.0712	0.27	0.10	0.0113
IL-6 (pg/mL)	7.83	4.45	0.001	8.97	5.73	0.0001
IL-10 (pg/mL)	4.45	2.21	0.2569	6.73	3.65	0.0769

hsCRP—high sensitivity C-reactive protein; ESR—erythrocyte sedimentation rate; TNF—tumor necrosis factor; IL-6—Interleukin-6; IL-10—Interleukin-10.

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
