# Peer review of "High Dose Intravenous Fish Oil Reduces Inflammation—A Retrospective Tale from Two Centers"

_nutrients, 2020, doi:10.3390/nu12092865_

Round 1

Reviewer 1 Report

I think this is a nice paper that adds observational data to the growing body evidence of fish oil using in PN. It is an important topic and has important implications for our PN patients.  I would like to see some additional data not reported to better understand the patients enrolled in the study for the paper to have been clinical applicability. 

Reviewer 2 Report

Nutrients-908748

High dose intravenous fish oil reduces inflammation 3 – a tale from two centers.

Review

This is a retrospective cohort study on the impact in inflammation and liver function tests of the addition of pure fish oil intravenous lipid emulsion in high dose as part of PN already containing first- or second-generation lipid emulsion. Two cohorts were studied, one with short term PN (IBD, subdivided in UC and CD) and another with long term PN (CIF).

The approach of the article seems interesting initially and in proper order. However, in my opinion, some points have to be addressed before publication:

1 - pag 1 – Title – I must admit that the title was eye-catching but I suggest a more informative one indicating the type of study (retrospective).

2 - pag 1 - Abstract – lin 29-30 – Since, total bilirubin concentration and SGOT did not reach significance, I will cut the sentence to something like “bilirubin concentration and SGOT did not varied over time”. Please, rewrite Conclusions. This is, in my opinion, an hypothesis generation study and it cannot be concluded that “pure fish oil ILE contributes to the reduction of inflammation…”

3 - pag 2 – lin 75 – Possibly “months” has to be changed to “month”.

4 - pag. 3 – lin 108-109 – “Nutrient supply… 0.15-0.2 g nitrogen/kg body weight/day”. Please provide the dose of protein instead of the dose of nitrogen. It is more informative.

5 – pag 3 -4 – Methods – Many variables were collected, but they were not reflected in the Results section in any way, i.e.: sodium, potassium, chloride, …creatinine, urea, glucose, albumin, platelets,… From “clinical variables”, cholestasis or AE were not present in Results.

6 – pag 4 - Statistical analysis – In my opinion, this section has to be re-written completely and change the results accordingly. The authors indicated that the test performed were Kruskal Wallis test for non-normally distributed variables and ANOVA with post hoc analysis for normally distributed variables. To my knowledge, these tests are used to compare variables in more than 2 groups. However, in the article only comparisons of variables pre- and post FO were presented for each group. No comparison amongst three groups were presented. Possibly the appropriate tests would be the Wilcoxon signed-rank test or the paired Student's t test. I suggest the first one, Wilcoxon signed-rank test, since the number or patients in each group is <30 and a non-parametric test seems more appropriated.

7 – pag 5 & 6 – Results – I understand that brevity is an advantage, but, in my opinion, the results presented need to be expanded. For example: which was the actual dose of glucose & fat per kg of weight? What was the dose of soybean fat emulsion per kg of weight? How evolved some nutritional markers as albumin or total protein and triglycerides? Blood glucose during the study? Hyperglycemia initially? Any sepsis episode during the study? Antibiotics especially in the IBD group? Corticosteroids, anti-TNF or anti IL-6 drugs as IBD treatment? I think all these variables could influence positively or negatively the results. Unification of variables in both tables will be more informative (i.e.: liver variables, inflammation related variables).

8 – pag 5 – Table 2 – Units of the variables would be of help.

9 – CRP and hsCRP – In Methodology (pag 3, lin 119) appeared hsCRP, in table 1 appeared C-reactive protein, and in table 2 again hsCRP. Please, homogenize and explain the abbreviation the first time it appeared. In addition, to my knowledge, hsCRP is a biomarker for low level of inflammation mostly in chronic diseases, but in the IBD group the levels of this variable were initially >>10 mg/dL? Please, explain briefly the use of this marker.

10 – pag 7 – lines 232-236 – “Nonetheless, we were able to demonstrate beneficial effects… “. Please include something like “our patients “ or “in the patients of our study” since to demonstrate the statement a randomized controlled trial should be necessary.

Other minor comments could be made, but I think important changes in Methodology and Results has to be made first as commented before.